# Effect of Metformin on Sertoli Cell Fatty Acid Metabolism and Blood–Testis Barrier Formation

**DOI:** 10.3390/biology13050330

**Published:** 2024-05-09

**Authors:** Gustavo Marcelo Rindone, Marina Ercilia Dasso, Cecilia Lucia Centola, Cristian Marcelo Sobarzo, María Noel Galardo, Silvina Beatriz Meroni, María Fernanda Riera

**Affiliations:** 1Centro de Investigaciones Endocrinológicas “Dr. César Bergadá” (CEDIE), Consejo Nacional de Investigaciones Científicas y Técnicas (CONICET), FEI–División de Endocrinología, Hospital de Niños Ricardo Gutiérrez, Buenos Aires C1425EFD, Argentina; grindone@cedie.org.ar (G.M.R.); mdasso@cedie.org.ar (M.E.D.); ccentola@cedie.org.ar (C.L.C.); mngalardo@cedie.org.ar (M.N.G.); smeroni@cedie.org.ar (S.B.M.); 2Instituto de Investigaciones Biomédicas (INBIOMED), Consejo Nacional de Investigaciones Científicas y Técnicas (CONICET), Facultad de Medicina, Universidad de Buenos Aires (UBA), Buenos Aires C1121ABG, Argentina; ciruba02@hotmail.com

**Keywords:** Sertoli cell, metformin, lipid metabolism, ketone bodies, blood–testis barrier, daily sperm production

## Abstract

**Simple Summary:**

Type 2 diabetes mellitus is a chronic disease afflicting millions of people worldwide. Though metformin is the main pharmacological treatment in children, little is known about the effects of this drug on testicular maturation, which takes place at this stage of life. Sertoli cells (SCs) have a central role in spermatogenesis since they support germ cell development. This is mainly achieved through blood–testis barrier (BTB) formation and the influence of SC metabolites on germ cells. We observed in SC cultures that metformin modifies fatty acid metabolism, ketone body production, and junction formation between SCs. Also, metformin treatment of juvenile rats provokes a mild but significant increase in BTB permeability. However, metformin does not affect testicular histology or meiosis progression. Moreover, BTB function is recovered, and daily sperm production is not affected in adult animals treated with metformin during the juvenile stage, suggesting that the effects on BTB permeability are reversible. Diabetes mellitus incidence is rising at staggering rates in children, and so is the exposure to metformin. Considering that a complete evaluation of the testicular function in children is usually not achievable, the present study seeks to add information about the safety of metformin treatment during this period of life.

**Abstract:**

Sertoli cells (SCs) are essential to maintaining germ cell development. Metformin, the main pharmacologic treatment for pediatric type 2 diabetes, is administered to children during SC maturation. The present study aimed to analyze whether metformin affects SC energy metabolism and blood–testis barrier (BTB) integrity. Primary SC cultures were used for the in vitro studies. In vivo effects were studied in Sprague–Dawley rats treated with 200 mg/kg metformin from Pnd14 to Pnd30. Metformin decreased fatty acid oxidation and increased 3-hydroxybutyrate production in vitro. Moreover, it decreased the transepithelial electrical resistance across the monolayer and induced ZO-1 redistribution, suggesting an alteration of cell junctions. In vivo, a mild but significant increase in BTB permeability and ZO-1 expression was observed in the metformin group, without changes in testicular histology and meiosis progression. Additionally, adult rats that received metformin treatment during the juvenile period showed no alteration in BTB permeability or daily sperm production. In conclusion, metformin exposure may affect BTB permeability in juvenile rats, but this seems not to influence spermatogenesis progression. Considering the results obtained in adult animals, it is possible to speculate that metformin treatment during the juvenile period does not affect testicular function in adulthood.

## 1. Introduction

Sertoli cells (SCs) play a central role in spermatogenesis since they provide nutritional and structural support to germ cells (GCs) within the seminiferous tubules. This is mainly achieved through the formation of the blood–testis barrier (BTB) and the provision of SC metabolites to GCs. The BTB is a complex dynamic structure established between adjacent SCs that divides the seminiferous epithelium into the basal and the adluminal compartments. This barrier isolates GCs as they progress through spermatogenesis and defines a microenvironment for their development [1]. It has been reported that if BTB integrity is not preserved, spermatogenesis and reproductive capacity are impaired [2,3,4].

GCs isolated in the adluminal compartment depend on the nutritional support of SCs, which display a versatile metabolic profile and supply lactate to GCs as an energy source [5,6,7]. As glucose is mainly directed to lactate production, SCs rely on complete fatty acid (FA) oxidation for their own energy supply [8,9]. In fact, in the absence of glucose, SCs maintain their viability for at least 48 h [10] and promote FA mobilization from lipid droplets [11], which highlights the importance of FA as an energy source. SCs also produce ketone bodies from FA, since they express the mitochondrial isoform of 3-hydroxy-3-methylglutaryl-CoA synthase (HMGCS2), a key enzyme for ketogenesis [8,12]. Ketone bodies are also suggested as energy substrates for GCs and spermatozoa [13,14]. 

SC functions, some of which are vital for spermatogenesis progression (BTB formation and synthesis of nutritional factors), are attained through a maturation process that begins at the end of the proliferative phase, around postnatal day (Pnd) 15 in rats [15]. Specifically, BTB establishment occurs in rats between Pnd25 and Pnd30 [16]. Human SCs undergo the last proliferative stage during the peripubertal period and achieve maturity at puberty [17,18].

Type 2 diabetes mellitus (T2DM) is a chronic disease afflicting millions of people worldwide. Patients often require pharmacological strategies to stabilize glycemia and avoid long-term complications. The effect of antidiabetic therapies on male reproduction has caused growing concern during the past few years, and many studies have been conducted to assess their potential risks [19]. Moreover, T2DM prevalence in children and adolescents is rising [20], which leads to young patients being exposed to antidiabetic agents during their development [21,22]. Metformin is one of the US Food and Drug Administration-approved pharmacological agents to treat this disease in pediatric populations and is the mainstay of treatment together with diet and exercise [23]. 

Considering the central role of SCs in male reproduction, unveiling the effect of metformin on this cell population is of key importance. In this line, several authors have analyzed the effects of metformin on lactate metabolism [24,25] and BTB integrity [26], but these studies concern the adult population, and they do not assess the effects of metformin administration during the period when BTB formation takes place. 

In the present study, we investigated the effects of metformin exposure during the critical period of SC maturation on lipid metabolism and BTB integrity. In addition, we analyzed the potential impact of juvenile exposure to metformin on BTB function and spermatogenic capacity in adult animals.

## 2. Materials and Methods

### 2.1. Materials

Metformin (1,1-dimethylbiguanide hydrochloride) was purchased from Sigma-Aldrich (St. Louis, MO, USA). The [9,10(n)-^3^H] palmitic acid was purchased from NEN (Boston, MA, USA). Phosphorylated ACC (Ser79), phosphorylated p70S6K (Thr389), total ACC, and total p70S6K antibodies were acquired from Cell Signaling Technology (Danvers, MA, USA). The androgen receptor (AR) antibody was purchased from Santa Cruz Biotechnology, Inc. (Dallas, TX, USA), claudin 11 antibody was obtained from Invitrogen (Thermo Fisher Scientific, Waltham, MA, USA), ZO-1 antibody was acquired from Zymed Laboratories (Thermo Fisher Scientific), and β-tubulin antibody was purchased from Sigma-Aldrich. Tissue culture media and all other drugs were acquired from Sigma-Aldrich. Sprague–Dawley rats (Rattus norvegicus) were obtained from the Central Animal Facility of the Department of Veterinary Sciences (Universidad de Buenos Aires, Argentina).

### 2.2. In Vitro Assays

#### 2.2.1. Sertoli Cell Isolation and Culture

SC isolation from testes of twenty-day-old male rats was performed as described in Riera et al. (2009) [10]. Isolated SCs were resuspended in culture medium: a mixture of Ham’s F-12 and DMEM (1:1; *v*/*v*) with 2.5 μg/mL amphotericin B, 10 μg/mL transferrin, 10 mM HEPES, 100 IU/mL penicillin, 1.2 mg/mL sodium bicarbonate, 5 μg/mL insulin, 5 μg/mL vitamin E, and 4 ng/mL hydrocortisone. SCs were cultured on 6- or 24-multiwell plates (5 μg DNA/cm^2^) or on Matrigel-coated inserts (Millicell HA inserts, Millipore; 15 μg DNA/cm^2^) placed on 24-multiwell plates at 34 °C in a mixture of 5% CO_2_:95% *v*/*v* air. Immunohistochemistry for SOX-9 (SC marker) and smooth muscle α actin (myoid cell marker) of SC cultures revealed that the remaining cell contaminants were of germ cell origin and represented less than 5% of the cells after 48 h in culture.

#### 2.2.2. Culture Conditions

After seeding, SC attachment was allowed for 48 h in the presence of insulin (5 μg/mL). Then, fresh medium without insulin was added. To measure ketone body production, fresh medium containing palmitate (100 μM) was used. Cells were maintained under basal conditions or stimulated with metformin (10 mM) for different periods. Cells incubated for 48 or 72 h were used to evaluate cell viability. Cells treated for 24 or 48 h were used to evaluate FA oxidation. Cells incubated for 24, 48, or 72 h were used for RT-qPCR analysis. For Western blot studies, cells treated for 2, 6, 24, 48, or 72 h were used. The 72 h-conditioned media were used to measure 3-hydroxybutyrate production.

To quantify transepithelial electrical resistance (TER), SCs were seeded on Matrigel-coated inserts. On day 3, testosterone (1 μM) was added to facilitate tight junction formation (control). On the same day, metformin (10 mM) was added and TER across the SC monolayer was recorded every 24 h until day 8. Cells cultured on 6-multiwell plates incubated for 24 or 72 h with testosterone in the absence or presence of metformin were used to analyze BTB-related gene expression.

#### 2.2.3. Fatty Acid Oxidation Assay 

FA oxidation assay was performed measuring the release of ^3^H_2_O into the incubation medium from SC cultured with [^3^H]-palmitate as previously described [27]. Results are expressed as pmol of palmitic acid/h/μg DNA.

#### 2.2.4. Determination of 3-Hydroxybutyrate Production

The 3-hydroxybutyrate was measured using the RANBUT reagent (Randox Laboratories Limited, Crumlin, UK) as previously described [12]. Results are expressed as nmol of 3-hydroxybutyrate/μg DNA.

#### 2.2.5. BTB Integrity Assay: Transepithelial Electrical Resistance (TER) Measurement

SC junction barrier establishment was assessed daily from day 4 to day 8 through the measurement of the TER across the SC monolayer as previously described [28]. Briefly, we measured the electrical resistance (ER) created by the epithelial monolayer against a short (~2 s) electrical current of 20 μA that circulated through 2 silver-silver chloride electrodes, placed in each compartment of the bicameral system. To calculate the area of resistance (ohms.cm^2^), ER was multiplicated by the area of the bicameral chamber. The net TER value was then obtained by subtracting the background, which was determined in inserts coated with Matrigel in the absence of SCs. Each time point had quadruplicate bicameral units. This experiment was performed three times on independent batches of cells.

#### 2.2.6. Real-Time PCR (RT-qPCR)

Gene expression was evaluated by RT-qPCR. For that purpose, total RNA was isolated from decapsulated testes or SC cultured in 6-multiwell plates with TRI Reagent (Sigma-Aldrich). RNA quantity was determined by spectrophotometry at 260 nm. Reverse transcription and real-time PCR were performed as previously described [28]. The specific primers used for RT-qPCR were previously described [12,28,29,30]. Amplification was performed as recommended by the manufacturer using the SYBR Green PCR Master mix (Applied Biosystems, Warrington, UK) and the Step One Real-Time PCR System (Applied Biosystems). To determine PCR product specificity, melting curves were acquired after amplification. The relative standard curve method was used to calculate relative gene expression. Data were normalized to β actin and β-2 microglobulin. 

#### 2.2.7. Western Blot Analysis 

Western blot analysis was carried out as previously described [10]. Antibodies against phosphorylated ACC (Ser79) and p70S6K (Thr389), total p70S6K, rat liver CPT1, androgen receptor (AR), and β-tubulin were used. A 1:4000 (CPT1) or 1:1000, dilution of primary antibodies was used. For chemiluminescence detection, a commercial kit from Cell Signaling Technology was used. Band intensities were estimated by densitometry scanning using the NIH Image Software (Scion Image 4.0.3.2, Scion Corporation, Frederick, MD, USA). Total p70S6K or β-tubulin levels were used as the loading control. 

#### 2.2.8. Cell Viability Test

Cell viability was assessed in SC cultured in 96-multiwell plates using the CellTiter 96^®^ AQueous Non-Radioactive Cell Proliferation Assay (Promega Corporation, Madison, WI, USA). 

#### 2.2.9. Immunofluorescent (IF) Detection of Claudin 11 and ZO-1 Protein

IF detection was carried out as previously described [28,30]. Briefly, SC monolayers were fixed, permeabilized, and blocked. For claudin 11 detection, the coverslips were incubated with a 1:50 dilution of antibody against claudin 11 in PBS overnight at 4 °C. Then, the coverslips were incubated with an anti-rabbit IgG fluorescein isothiocyanate (FITC) conjugate (1:25; Vector Laboratories, Burlingame, CA, USA). For ZO-1 detection, the coverslips were incubated with a 1:50 dilution of the primary FICT-conjugated antibody against ZO-1 in PBS overnight at 4 °C. After washing, coverslips were incubated with a 1:100 dilution of a biotinylated secondary antibody anti-FITC. Subsequently, they were incubated with Neutralite avidin-FITC conjugate (Southern Biotech, Birmingham, AL, USA) diluted to 1:200. Nuclei were counterstained with DAPI (Invitrogen; Thermo Fisher Scientific, Inc., Waltham, MA, USA). For negative controls, primary antibodies were replaced by PBS. Finally, coverslips were mounted and observed using an Axiophot fluorescent microscope with epi-illumination (Carl Zeiss Inc., Oberkochen, Germany).

### 2.3. In Vivo Assays

#### 2.3.1. Animals

Pregnant Sprague–Dawley rats were housed in individual cages under controlled conditions of temperature and humidity, 12 h light/dark cycles, and free access to water and commercial pellet laboratory chow (Rat-Mouse Diet, Asociación de Cooperativas Argentina, Buenos Aires, Argentina). Offspring delivery day was considered postnatal day 0 (Pnd0). All procedures were conducted in agreement with the international guidelines and regulations of the NIH and approved by the Institutional Committee for the Care and Use of Laboratory Animals from the Hospital de Niños “Dr. Ricardo Gutiérrez” (Resolution No. 2017/007). 

#### 2.3.2. Experimental Design 

At Pnd3, pups were sexed according to their anogenital distance, and litters of a maximum of ten pups were left per mother whenever possible. At Pnd15, male pups were weighed and assigned to one of two groups: control and metformin. Randomization was carried out so that there was no weight difference between groups at the start of treatment. The control group received vehicle (sterile saline solution) and the metformin group (MET) received an oral dose of 200 mg/kg/day. Dosing criteria were based on prior studies showing that, in rodents, a 200 mg/kg dose of metformin is required to obtain similar plasma concentrations to those observed in human patients treated with this drug [31]. Dose translation was performed using the body surface area calculation method [32]. The animals were treated from Pnd15 to Pnd30. At Pnd31, the animals were euthanized by CO_2_ asphyxiation; the testes were removed and used for histological and TUNEL analysis, BTB permeability assessment, intratesticular testosterone determination, and RNA extraction. A set of animals was treated from Pnd15 to Pnd30 with vehicle or 200 mg/kg/day of MET and maintained without further treatment until Pnd90. Testes were used for histological analysis, BTB permeability assay, and daily sperm production determination in Pnd90.

#### 2.3.3. Histological Analysis and Meiosis Progression 

Testes were fixed in Bouin solution and embedded in paraffin (n = 5/group). Sections (3–5 μm) obtained from both poles and equatorial areas were stained with hematoxylin and eosin. The seminiferous tubule morphology was examined by light microscopy. For meiosis progression analysis, seminiferous tubule cross-sections (500 per animal) from the control and MET groups were assigned to three different categories. Each tubule was classified according to the most differentiated GC stage it contained. Exclusion criteria between categories were the presence of round spermatids (RSs), the absence of RS, and the presence of pachytene spermatocytes (PSs) or tubules lacking both RSs and PSs, with the presence of preleptotene, leptotene, or zygotene spermatocytes or spermatogonia (non-RSs or PSs). 

#### 2.3.4. Blood–Testis Barrier Integrity Assay

The permeability of the BTB was assessed using a biotin tracer as described by Gorga et al. (2021) [29]. Six testes from each group were used. Briefly, testes were injected with the biotin tracer (EZ-Link Sulfo-NHS-LC-Biotin; MW: 0.6 kDa, Pierce Biotechnology, Rockford, IL, USA) and incubated for 30 min at 34 °C. Then, testes were fixed in 4% paraformaldehyde and embedded in paraffin. Sections (3 μm) obtained from the testis poles and equatorial areas were deparaffinized and hydrated. For detection, sections were incubated with streptavidin-Alexa Fluor 555 (1:3000, Invitrogen). Cell nuclei were stained with DAPI. Sections were observed using an Axiophot fluorescence microscope with epi-illumination (Zeiss, Germany). Nonconsecutive testis sections (n = 3) were examined under the microscopy and 400 seminiferous tubule sections were evaluated and counted. Results are expressed as the percentage of permeated tubules over total ones. 

#### 2.3.5. Intratesticular Testosterone Determination

At Pnd31, testes from both groups were used to evaluate intratesticular testosterone (n = 6/group). Testosterone was extracted from 150 mg of testis homogenates with diethyl ether followed by evaporation of the organic phase and reconstitution of extracted testosterone in 0.1% PBS. Testosterone concentration was measured as previously described [33]. Results are expressed as ng testosterone/g testis.

#### 2.3.6. Apoptosis Determination by TUNEL

At Pnd31, testes from control and MET groups (n = 4/group) were fixed in formalin 10% and embedded in paraffin. Two nonconsecutive testis sections (3–5 μm) were blocked with a horse serum dilution. Terminal deoxynucleotidyl transferase dUTP nick-end labeling (TUNEL) assay was performed as previously described [33]. The In Situ Cell Death Kit (Roche Applied Science, Indianapolis, IN, USA) was used. For negative controls, the TUNEL reaction mixture without terminal transferase was used. Cell nuclei were stained with DAPI. Epifluorescence microscopy was performed using an Axio Imager 2 microscope (Zeiss, Germany) coupled with an Axiocam 202 monochrome camera (Zeiss). Histological sections were counted and averaged per animal. A minimum of 400 transversal seminiferous tubules were selected in each tissue section. The number of TUNEL-positive cells per round seminiferous tubule was quantified in each tissue section using Image J version 1.53k software (NIH, Bethesda, MD, USA). The incidence of apoptosis was also reported as the apoptotic index (AI). The AI was calculated as the percentage of seminiferous tubules showing three or more TUNEL-positive cells. 

#### 2.3.7. Daily Sperm Production (DSP)

At Pnd90, testes were collected and weighed (n = 4/group). They were processed according to the procedure previously described [33]. The parenchyma was homogenized, and the homogenates were sonicated. Samples were diluted, placed into a Neubauer chamber, and counted in quadruplicate. Elongated spermatid nuclei with the characteristic shape of step 19 spermatids and resistant to homogenization were counted to determine the number of spermatid nuclei. DSP was calculated by dividing the number of spermatids by 6.1, which is the number of days of the seminiferous cycle in which these spermatids are present in the seminiferous epithelium.

### 2.4. Statistical Analysis 

Normality (Shapiro–Wilk test) and homoscedasticity (Levene test) were evaluated in each data set. One-way ANOVA and the post hoc Tukey–Krämer multiple-comparisons test were performed using InfoStat version 2020 (Grupo InfoStat, FCA, Universidad Nacional de Córdoba, Córdoba, Argentina). A *p*-value < 0.05 was considered statistically significant.

## 3. Results and Discussion

### 3.1. Metformin Affects FA Oxidation and Ketone Body Production in SC Cultures

In the first set of experiments, we assessed the potential effect of metformin on FA oxidation in SCs, which is their main strategy to obtain energy. For this purpose, we evaluated ^3^H-palmitate oxidation through the measurement of metabolic H_2_O production in cultured SC isolated from 20-day-old rats incubated in the absence or presence of metformin (10 mM) for 24 or 48 h. Cell viability tests performed after the 24, 48, and 72 h incubation periods showed that metformin (10 mM) did not affect cell viability (Table 1). Figure 1a shows that metformin inhibited FA oxidation at 24 h and 48 h. This decrease in FA oxidation agrees with prior findings showing that metformin reduced palmitate oxidation in L6–C11 skeletal muscle cell cultures [34]. It is also known that metformin acts as an inhibitor of mitochondrial respiratory chain complex I activity [35]. In this respect, metformin and rotenone (another complex I inhibitor) have been shown to reduce FA oxidation, which led to lipid accumulation in MCF7 and MDA-MB-468 breast cancer cell lines [36]. These latter observations and our results might imply that metformin decreases complete FA oxidation in SCs through the inhibition of the respiratory chain complex I activity. 

To be oxidized, FAs bound to coenzyme A (acyl-CoA) are transported by the enzymes carnitine palmitoyl transferase (CPT) 1 and 2. The transport through CPT1 is a critical regulatory step of acyl-CoA entry into the mitochondria and also into FA oxidation. Once inside the organelle, the long- and medium-chain acyl-CoA dehydrogenases, LCAD and MCAD, respectively, catalyze the first step in FA oxidation. After several reactions, acetyl-CoA is produced and directed to the Krebs cycle to generate reduction equivalents, which are ultimately conducted through the respiratory chain to produce ATP. To determine whether some of these factors could be involved in the observed decrease in FA metabolism in response to metformin, we measured *Cpt1*, *Mcad*, and *Lcad* mRNA levels. Figure 1b shows that metformin reduced *Cpt1* and *Mcad* mRNA levels, while no changes were observed in *Lcad* mRNA levels. This figure also shows that metformin significantly reduced CPT1 protein levels after 72 h treatment. In agreement with these results, previous studies have demonstrated that metformin decreases *Cpt1* expression in freshly isolated hepatocytes [37]. In addition, in the high-fat-diet model of induced insulin resistance, metformin-treated animals also showed decreased *Cpt1* expression in skeletal muscle cells [38]. All these results may indicate that the observed decrease in FA metabolism can be partly explained by changes in gene and protein expression. Considering that FAs are the main source of energy for SCs, our results also suggest that metformin may be affecting SC metabolism. 

As mentioned before, it is widely accepted that metformin transiently inhibits mitochondrial complex I, reducing the respiratory chain energy output. This leads to an increase in the AMP/ATP ratio, ultimately activating the AMPK, a critical sensor of cellular energy homeostasis [39]. The mammalian target of rapamycin complex 1 (mTORC1)/p70SK6 signaling pathway is an AMPK target and functions as an intracellular nutrient sensor that controls protein synthesis and cellular energy homeostasis. Since it has been shown that metformin regulates this pathway in other cell types [40,41], we evaluated AMPK and mTORC1 activation by measuring the levels of phosphorylated acetyl-CoA carboxylase (P-ACC) and phosphorylated p70S6K (P-p70SK6), respectively. SCs were incubated for different periods—2, 6, 24, and 48 h—with metformin (10 mM). Figure 2a shows that metformin increased P-ACC levels and decreased P-p70S6K levels at all the incubation periods tested. These results suggest that metformin activates AMPK and inhibits mTORC1/p70S6K pathways in SCs.

As mentioned above, CPT1 catalyzes the rate-limiting step in FA oxidation, and its activity is allosterically inhibited by malonyl-CoA. Phosphorylated ACC by AMPK, which exhibits decreased activity, generates lower malonyl-CoA levels, in turn increasing CPT1 activity [42]. Thus, AMPK activation will increase acyl-CoA influx into the mitochondria for its oxidation. Considering (a) that SCs produce ketone bodies from FA, (b) that ketogenesis takes place when acetyl-CoA levels exceed the capacity of the mitochondria to oxidize it, and (c) that metformin increases AMPK activity and decreases the production of ^3^H_2_O from ^3^H-palmitate, we hypothesized that FA influx into the mitochondria is indeed favored, and we considered the possibility of acetyl-CoA deviation towards ketone body production. SC cultures were incubated with metformin (10 mM) for 72 h. Figure 2b shows that metformin increases 3-hydroxybutyrate production in SCs. It has also been shown that metformin increases 3-hydroxybutyrate production in isolated hepatocytes [38]. The enzyme 3-hydroxymethylglutaryl-CoA synthase 2 (HMGCS2), responsible for the synthesis of hydroxy-methylglutharyl-CoA, and monocarboxylate transporter 4 (MCT4), which in turn exports ketone bodies from the cell, are important regulatory steps in ketone body synthesis and secretion. Hence, we studied whether metformin also modifies their expression. Figure 2b shows that metformin markedly increased *Hmgcs2* mRNA levels after 72 h treatment, while *Mct4* mRNA levels showed an earlier increase after 24 and 48 h. At this point, it is important to consider that the conditions leading to an increase in plasma ketone bodies, such as starvation or diabetes, correlate with increased hepatic *Hmgcs2* mRNA levels [43,44]. A similar response was observed in SCs, in which the coincubation with GCs increased 3-hydroxybutyrate production and *Hmgcs2* expression [12]. Once ketone bodies are synthesized, export regulation by MCTs may also contribute to the regulation of ketone body secretion. MCT4 may be responsible for the exit of ketone bodies from the cells due to its lower affinity for monocarboxylate than other MCTs [45]. In agreement with this, we have demonstrated that GCs and basic fibroblast growth factor (bFGF) upregulate *Mct4* expression in SCs as a mechanism to increase 3-hydroxybutyrate secretion [12]. Our results agree with those obtained by Nna et al. (2021) [25], who observed an increase in testicular *Mct4* expression in response to metformin treatment of adult diabetic rats. Altogether, we postulate that metformin might regulate the entry of FA into the mitochondria through changes in ACC phosphorylation and the expression of key proteins—HMGCS2 and MCT4—as molecular mechanisms to increase the secretion of ketone bodies by SCs. 

Adequate GC nutrition is a key factor for successful spermatogenesis. Spermatocytes and spermatids are unable to metabolize glucose and require the lactate produced by SCs [6,7]. Apart from lactate, Koga et al. (2000) [13] characterized a testicular isoform of succinyl-CoA transferase (SCOT-t) in post-meiotic GCs, which suggests that ketone bodies may also contribute to GC energy homeostasis. Previous reports showing an impact of metformin on SC lactate production [24,46], and the present study, which describes an increase in ketone body production, support the hypothesis of metformin having a positive effect on energy availability for GCs. However, as these changes are at the expense of alterations in SC energy metabolism, particularly in the FA oxidation pathway, they should not be undoubtedly deemed as beneficial.

### 3.2. Metformin Alters the Formation of Cell Junctions in Cultured SCs and Increases BTB Permeability in Juvenile Male Rats

In addition to the nourishment provided to the germinal epithelium, SCs further contribute to spermatogenesis, creating a suitable microenvironment for successful meiotic division. SCs interact with each other by displaying every known cell–cell junction type: tight and gap junctions, desmosomes, and basal ectoplasmic specializations [1]. Through these interactions, they establish the BTB, which provides the aforementioned microenvironment. Considering the importance of the BTB in the physiology of the seminiferous tubules, we evaluated the effects of metformin on the integrity of this permeability barrier. To this end, we assessed the establishment of junctions between cultured SC daily from day 4 to day 8 by measuring transepithelial electrical resistance (TER) across the SC monolayer. Bearing in mind the role of androgens in the promotion of BTB formation, testosterone was added to SC cultures and considered a control condition. Figure 3a shows that metformin significantly reduced TER. This result suggests a disruptive effect of metformin on the junctions between SCs. We then explored the effects of metformin on mRNA levels of androgen receptor (*Ar*) and intercellular junction proteins such as occludin (*Ocln*), claudin 11 (*Cldn11*), ZO-1 (*Tjp1*), and connexin 43 (*Cx43*) in SC cultures. The AR is a ligand-dependent nuclear transcription factor responsible for androgen-mediated genomic actions. It has been observed that SC-specific AR conditional knockout mice have increased BTB permeability [47] and decreased expression of claudin 11, occludin, and several other proteins [48]. Both occludin and claudin 11 are tight-junction integral membrane proteins, while the ZO-1 is a scaffold protein that links occludin and claudin 11 to the actin cytoskeleton [49]. Connexin 43 is involved in forming gap junctions between neighboring SCs, and between SCs and GCs [50]. Claudin 11 null mice or conditional SC Cx43-knockdown mice show alterations in SC maturation and spermatogenesis [51,52]. Figure 3b shows that metformin decreased *Ar*, *Ocln*, *Clnd11*, *ZO-1*, and *Cx43* mRNA levels after 72 h incubations. 

Changes in the cellular distribution of junction proteins might be another mechanism triggering TER decrease. For this reason, we decided to evaluate claudin 11 and ZO-1 cellular localization in SC cultures by IF. Figure 4 shows that, in control conditions, claudin 11 and ZO-1 were localized in cell–cell contact surfaces, and that metformin promoted a redistribution of ZO-1 from the cell membrane to the cytoplasm but did not modify claudin 11 localization. In this respect, McCabe et al. (2016) [53] showed a clear correlation between SC tight-junction integrity and the expression of claudin 11 and occludin in vitro. In addition, it has been demonstrated that compounds that disrupt BTB integrity promote the relocation of ZO-1 from the cell–cell interface to the cytoplasm [54]. Considering all the evidence, the concurrent reduction in expression of *Ar* and other key proteins for SC junctions and the redistribution of ZO-1 seem like a plausible explanation for the observed changes in the TER in SC cultures. As these changes appear to be prejudicial for BTB integrity, we further evaluated whether these effects have any relevance in vivo. 

Cell cycle withdrawal is a required step for SC maturation. Several reports have shown that, at Pnd15, rat SC proliferation becomes barely detectable, at which point they begin to express intercellular junction molecules that allow for BTB establishment [15,16]. However, it is only by Pnd25-30 that the BTB is completely functional, which coincides with the final steps of meiosis II (around Pnd26 in rats) [16,55]. The following set of experiments was designed to analyze whether metformin treatment during the BTB formation period modifies its permeability. To this end, male rats received a daily oral dose of metformin (200 mg/kg; MET) or saline solution (control) from Pnd15 to Pnd30. No differences in weight gain were observed between MET and control animals. At Pnd31, body and testis weights were similar in both groups (Table 2). The diffusion of a biotin tracer from the interstitial to the adluminal compartment was evaluated to assess BTB integrity. Figure 5a shows representative pictures obtained from control and MET-treated animals. It can be noted that the tracer diffused into the adluminal compartment in the tubules with impaired BTB permeability, while it was excluded from this compartment in intact tubules. This figure also shows that a mild but significantly higher number of permeated tubules was observed in MET-treated animals. This result agrees with those obtained in vitro showing that metformin reduced TER values. Altogether, these results suggest that BTB functionality may be altered due to metformin exposure during its formation period. In this context, Ye et al. (2019) [26] showed that metformin reverses BTB disruption induced by a high-fat diet in adult mice. The discrepancy between the results obtained by Ye et al. (2019) [26] and ours may be attributed to the dissimilar animal models used: adult obese versus juvenile normal-weight animals. 

We then explored the mechanisms that could explain in vivo metformin effects on BTB permeability. First, as androgens are the main regulatory factors of BTB formation, intratesticular testosterone (ITT) and androgen receptor mRNA and protein levels were determined. Figure 5b shows that MET-treated animals exhibited similar ITT levels and higher *Ar* mRNA levels than control animals. However, AR protein levels were not modified in MET-treated animals. Secondly, the expression levels of intercellular junction proteins in response to metformin treatment were analyzed. Figure 5c shows that animals from the MET group showed increased *ZO-1* mRNA levels with no changes in *Ocln*, *Cldn11*, or *Cx43* mRNA levels. Therefore, the in vitro decrease in *Ar*, *Ocln*, *Cldn11*, *ZO-1*, and *Cx43* mRNA levels was not replicated in vivo. We hypothesize that the changes observed in vivo could be smaller (or absent) due to an enhanced adaptive response of SC to the effects of metformin in a physiological context. In addition, the increase in BTB permeability in vivo may not be attributable to a reduction in the expression of cell junction proteins. It is important to note that studies in SC-specific AR knockout mouse models have shown that androgens play a major role in regulating *Cldn11*, *Ocln*, and *ZO-1* expression [48,56]. Also, it has been demonstrated that testosterone increases *Cldn11* expression and induces claudin 11 and occludin recruitment between SCs [57]. Therefore, we postulate that the differential regulation of *Ar* expression by metformin between the in vitro and in vivo experiments may partly explain the differential effects on cell junction protein expression. Further studies will be necessary to determine how metformin alters BTB permeability in juvenile rats. 

Regarding ZO-1 function, it is classically known as an adaptor protein in tight junctions. In addition, ZO-1 is associated with gap and adherent junctions [58,59]. Our results showed an increase in *ZO-1* expression, along with increments in BTB permeability in the MET group. In this regard, claudin 11 knockout mice that showed alterations in BTB function also presented increased expression of ZO-1, as well as other cell junction-related genes [51,60]. Altogether, these results suggest that the upregulation of *ZO-1* expression may constitute a mechanism to compensate for increased tight-junction permeability.

As previously mentioned, BTB is essential for spermatogenesis progression. Therefore, changes in BTB dynamics would translate into disturbances in spermatogenesis [61,62]. With this information in mind, we performed a histological analysis of spermatogenesis progression in control and MET-treated animals. Figure 6a shows that seminiferous tubules had normal cellular associations and that SCs and GCs remained histologically unaltered in both groups. Figure 6b shows that, in both groups, there was a similar percentage of tubules with round spermatids (RSs), with pachytene spermatocytes (PSs), or without RSs or PSs but with the presence of germ cells in earlier stages of development (non-RSs or PSs), suggesting no changes in meiosis progression. In addition, we evaluated whether MET treatment affected cell apoptosis. Figure 6c shows a similar trend of apoptotic events in control and MET-treated animals considering the number of TUNEL-positive cells per seminiferous tubule or apoptotic index (AI, percentage of tubules that exhibit 3 ≥ TUNEL-positive nuclei). Many studies have shown that BTB disruption in adult animals leads to disturbances in seminiferous epithelium homeostasis and alterations in spermatogenesis [2,54,63]. In addition, an increase in cell apoptosis has been observed in adult animals with altered BTB permeability [64,65]. However, less is known about the impact of BTB disruption during its formation period on the seminiferous tubule architecture, the first spermatogenic wave, or testicular cell apoptosis. In this relation, claudin 11 and connexin 43 knockout mice showed altered seminiferous tubule morphology, increased cell apoptosis, and impaired spermatogenesis after the juvenile period [51,52]. In contrast, juvenile rats treated with a low dose of the herbicide glyphosate showed increased BTB permeability, but only a few seminiferous tubules with a disorganized epithelium [29]. The latter evidence suggests that spermatogenesis can progress with a permeable BTB in young animals and that it might be necessary to study the animals at a later stage of development to observe alterations in the seminiferous tubule. Nevertheless, given the mild nature of the observed changes in BTB permeability in vivo, together with the lack of effects on meiosis progression and cell apoptosis, it is possible that metformin actions on BTB integrity under the conditions tested might not be potent enough to affect ongoing spermatogenesis. 

Then, we evaluated the impact of metformin treatment during the juvenile period on testicular function in adult animals. For this purpose, animals were treated from Pnd15 to Pnd30 and then allowed to grow until Pnd90. Table 3 shows that no changes were observed in animal, testis, or epidydimal weight between groups. Figure 7a shows that adult animals treated with MET during the juvenile stage showed no differences in BTB permeability compared with control animals. Also, seminiferous tubules had normal cellular associations and architecture and showed complete spermatogenesis in both groups (Figure 7b). Finally, Figure 7c shows that daily sperm production was not modified by MET treatment. These results suggest that treatment with metformin during the juvenile period might not alter adult testicular function. The results obtained for histological and daily sperm production analysis are in contrast with those observed in mice exposed in utero to metformin. These animals showed an increase in the number of tubules with sloughing GCs in the lumen, a decrease in GC number per seminiferous tubule, and a lower number of spermatozoa in the epididymis in adult animals [66]. Moreover, neonatal rats treated with metformin during the SC proliferative period also showed a decrease in daily sperm production [33]. Therefore, these observations suggest that metformin has a different impact on testicular function depending on the developmental period during which animals are treated. 

## 4. Conclusions

The results obtained suggest that metformin alters SC lipid metabolism and tight-junction assembly in vitro. However, exposure to metformin during the BTB formation period in rats has minimal effects on BTB permeability and does not modify spermatogenesis progression. Additionally, adult animals treated during the juvenile stage showed preserved BTB function, without changes in daily sperm production. Altogether, the results obtained in this report suggest that treatment with metformin during the juvenile period does not induce long-term damage in testicular function. The incidence of T2D and obesity in children and adolescents is rising at staggering rates, as is exposure to metformin. Considering that a complete evaluation of testicular function in children is usually not achievable, the present study seeks to add information about the safety of metformin treatments during early stages of development.

## Figures and Tables

**Figure 1 biology-13-00330-f001:**
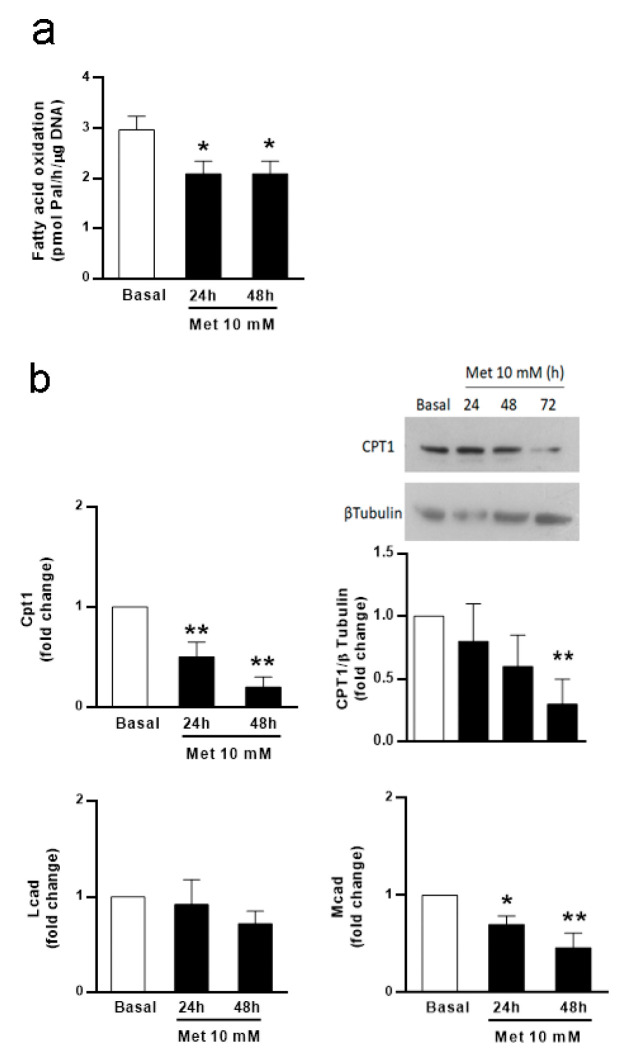
Effect of metformin on fatty acid oxidation; *Cpt1*, *Lcad*, and *Mcad* mRNA levels; and CPT1 protein levels in SC cultures. SC cultures were incubated in the absence (Basal) or presence of metformin (Met) 10 mM for variable periods (24, 48, and 72 h). (**a**) Measurement of ^3^H_2_O levels after incubating the cells with [^3^H] palmitic acid was carried out to assess fatty acid oxidation. Results are expressed as pmol of palmitic acid/h/μg DNA. (n = 3). * *p* < 0.05 vs. Basal. (**b**) Total RNA was extracted and RT-qPCR for *Cpt1*, *Lcad*, and *Mcad* was performed. Graphics show pooled data from four independent experiments, indicating fold change in mRNA levels relative to Basal. Results are expressed as mean ± S.D. * *p* < 0.05; ** *p* < 0.01 vs. Basal. Cell extracts were subjected to Western blot analysis using specific antibodies for CPT1 and β-tubulin. A representative immunoblot is shown together with data from three independent experiments indicating the fold variation in CPT1 (ratio of CPT1 to β-tubulin in each sample) relative to Basal. ** *p* < 0.01 vs. Basal.

**Figure 2 biology-13-00330-f002:**
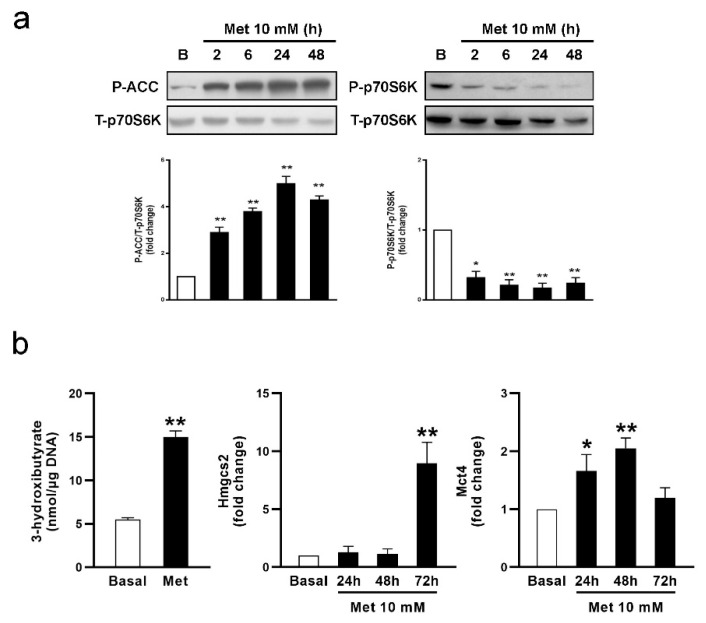
Effect of metformin on P-ACC and P-p70S6K levels, 3-hydroxybutyrate production, and *Hmgcs2* and *Mct4* mRNA levels in SC cultures. (**a**) SC monolayers were incubated in the absence (Basal; B) or presence of Met 10 mM for variable periods (2, 6, 24, and 48 h). Cell extracts were subjected to Western blot analysis using specific antibodies for P-ACC and P-p70S6K. Representative immunoblots are shown. Graphics show data from three independent experiments indicating the fold change in phosphorylation (ratio of P-ACC and P-p70S6K to T-p70S6K in each sample) relative to Basal. * *p* < 0.05; ** *p* < 0.01 vs. Basal. (**b**) SC monolayers were maintained in Basal conditions or incubated with Met 10 mM for variable periods (24, 48, and 72 h). 3-Hydroxybutyrate production was determined in 72 h-conditioned media. Data represent triplicate incubations in one representative experiment out of three (** *p* < 0.01 vs. Basal). Total RNA was extracted and RT-qPCR for *Hmgcs2* and *Mct4* was performed. Graphics show pooled data from four independent experiments indicating fold change in mRNA levels relative to Basal. Results are expressed as mean ± S.D. * *p* < 0.05; ** *p* < 0.01 vs. Basal.

**Figure 3 biology-13-00330-f003:**
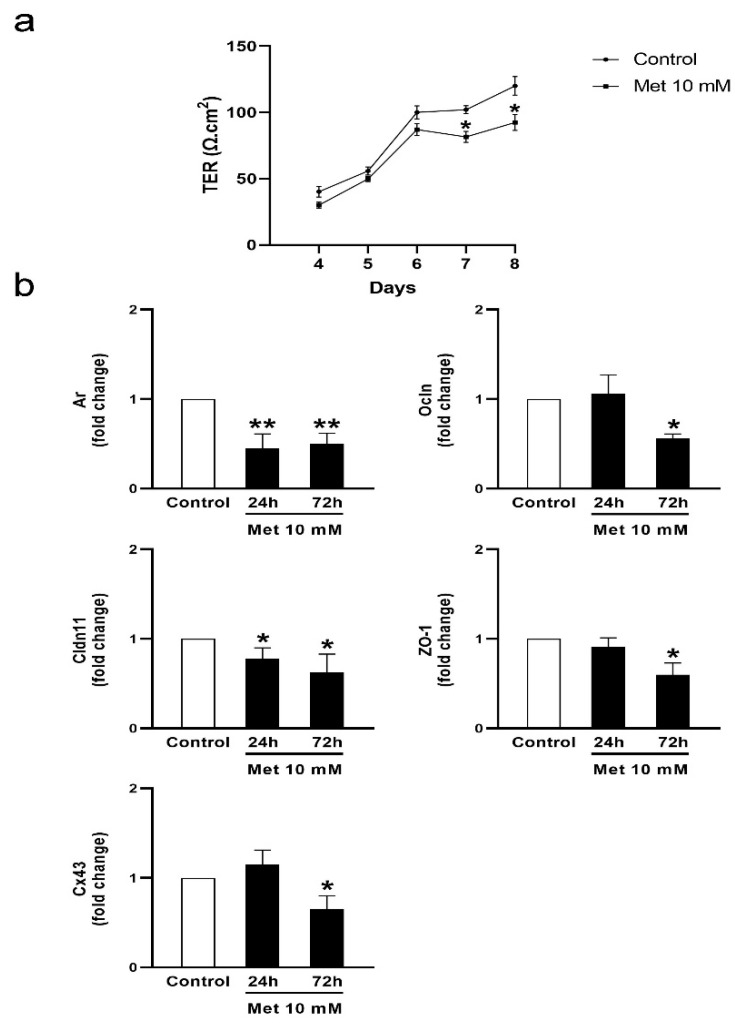
Effect of metformin on TER across SCs and mRNA levels of BTB-related proteins. (**a**) SC cultures were incubated in the presence of testosterone (1 μM) alone (control) or with the addition of Met 10 mM. TER was registered from days 4 to 8. Values represent mean ± S.D. of triplicate wells of one representative experiment out of three. Asterisks indicate statistical differences for each day, *p* < 0.05. vs. control. (**b**) SC monolayers were incubated in the presence of testosterone alone (control) or with the addition of Met 10 mM for 24 or 72 h. Then, total RNA was obtained and RT-qPCR for *Ar*, *Ocln*, *Cldn11*, *ZO-1*, and *Cx43* was performed. Graphics show pooled data from four independent experiments indicating fold change in mRNA levels relative to Basal. Results are expressed as mean ± S.D. * *p* < 0.05; ** *p* < 0.01 vs. Basal.

**Figure 4 biology-13-00330-f004:**
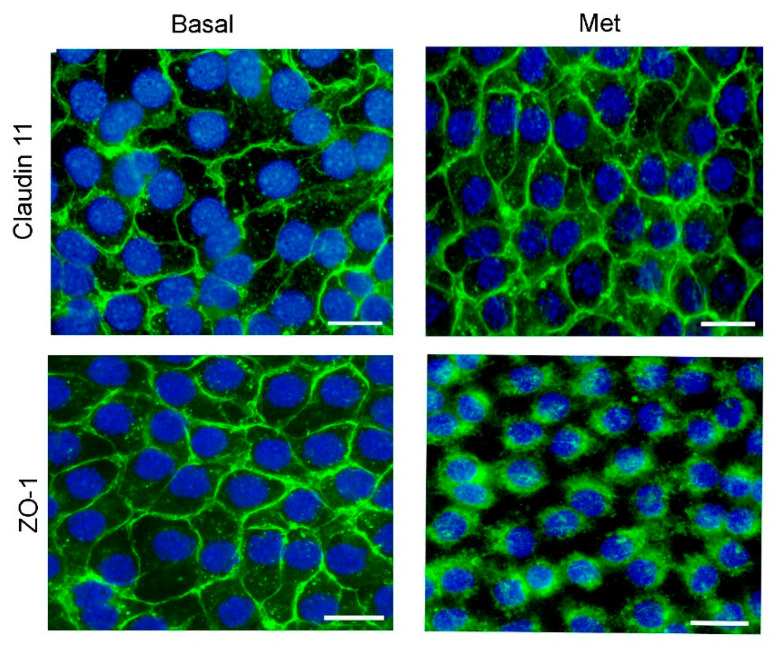
Effect of metformin on claudin11 and ZO-1 localization in SC cultures. SCs were maintained under Basal conditions or incubated with Met 10 mM for 24 h. Claudin11 and ZO-1 were revealed by immunofluorescence. Nuclei were stained with DAPI. Bars: 20 μm.

**Figure 5 biology-13-00330-f005:**
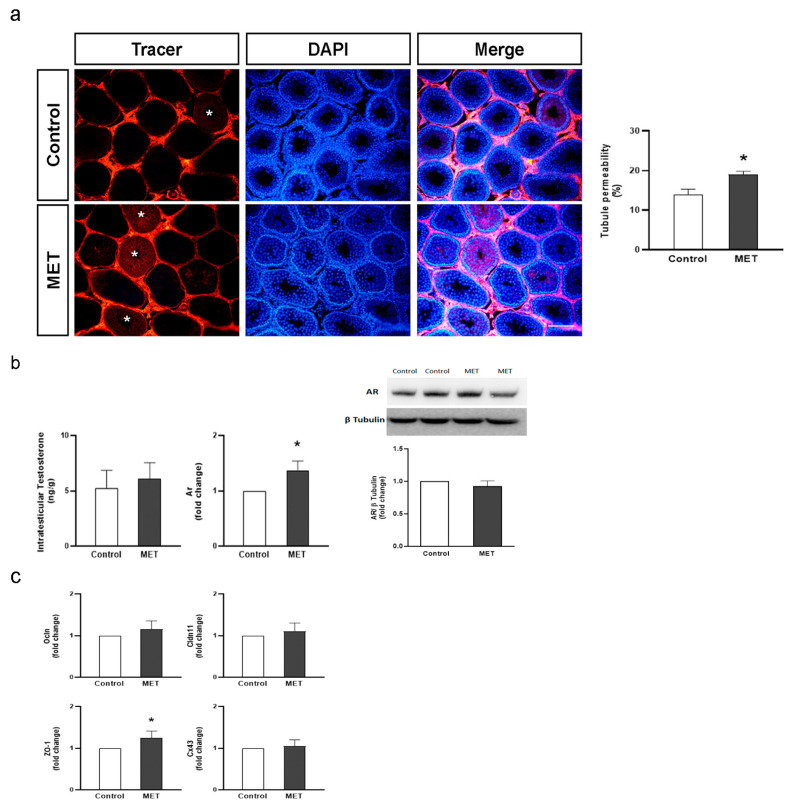
Effect of metformin treatment on BTB permeability, intratesticular testosterone levels, and *Ar* and BTB-related protein mRNA levels in juvenile rats. Animals (n = 6/group) from control and MET (200 mg/kg/day) groups were treated from Pnd14 to Pnd30. At Pnd 31, testes were removed. (**a**) To evaluate BTB permeability, testes were injected with a biotin tracer (red), and cell nuclei were dyed with DAPI (blue). Representative fields are shown in the left panels (scale bar, 100 µm). Permeated tubules were identified as those showing tracer presence in the adluminal compartment. White asterisks show permeated tubules. The quantification of permeable tubules was performed. Results are expressed as mean ± S.D. * *p* < 0.05 vs. control. (**b**) Testes were used to evaluate intratesticular testosterone and *Ar* mRNA levels by RT-qPCR. Additionally, testis extracts were used to evaluate AR and β-tubulin levels by Western blot analysis. Representative immunoblots show two representative comparisons between control and MET animals. The lower panels show pooled data from six animals indicating the fold variation in AR (ratio of AR to β-tubulin in each sample) relative to control. Results are expressed as mean ± S.D. * *p* < 0.05 vs. control (**c**). Total RNA was extracted from the whole testis and RT-qPCR was performed to measure mRNA levels of *Ocln*, *Cldn11*, *ZO-1*, and *Cx43*. Graphics show data indicating the fold variation in mRNA levels relative to control. Results are expressed as mean ± S.D. * *p* < 0.05 vs. control.

**Figure 6 biology-13-00330-f006:**
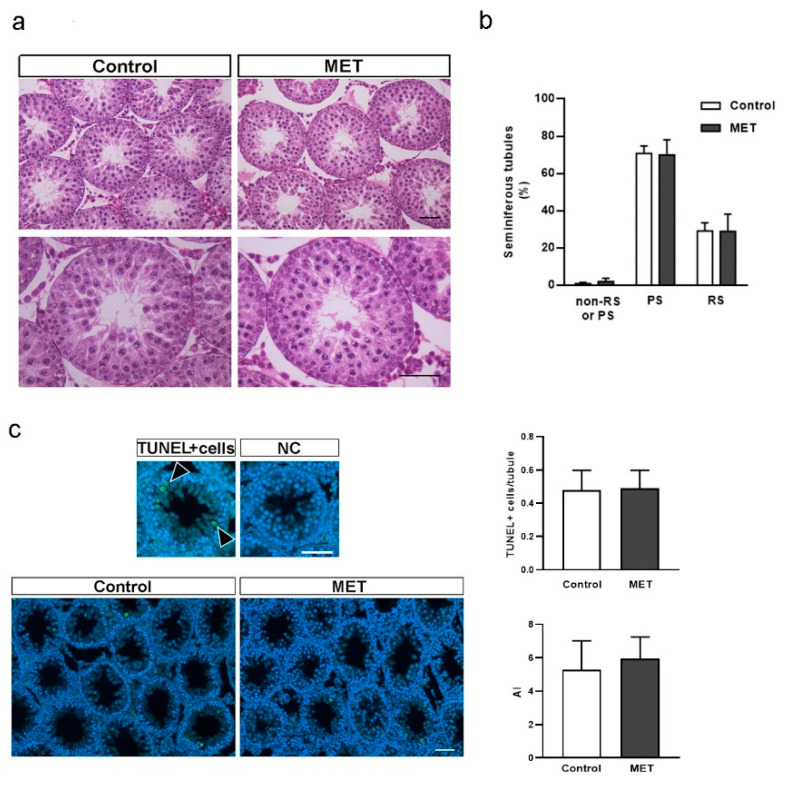
Effect of metformin treatment on testicular histology, meiosis progression, and apoptosis in juvenile rats. Animals from control and MET groups were treated from Pnd14 to Pnd30. At Pnd31, testes were removed and fixed in Bouin solution. Sections (3–5 µm) were stained with hematoxylin/eosin and examined by light microscopy. (**a**) Representative images of testis sections are shown. Higher magnification images are shown in the lower panels (scale bar, 100 μm). (**b**) Meiosis progression was assessed in sections obtained from the control and MET groups (n = 5/group). Seminiferous tubules were classified according to the most differentiated GC stage they contained (pachytene spermatocytes (PSs), round spermatids (RSs), or more immature GCs (non-RSs or PSs), and results are expressed as percentages. (**c**) Representative photomicrographs showing cross-section examples of a seminiferous tubule with three or more TUNEL-positive cells and a negative control (NC). Arrowheads indicate TUNEL-positive cells. Scale bar, 50 μm. Graphics show the number of TUNEL-positive cells per tubule and the apoptotic index (AI) (n = 4/group). Results are expressed as mean ± S.D.

**Figure 7 biology-13-00330-f007:**
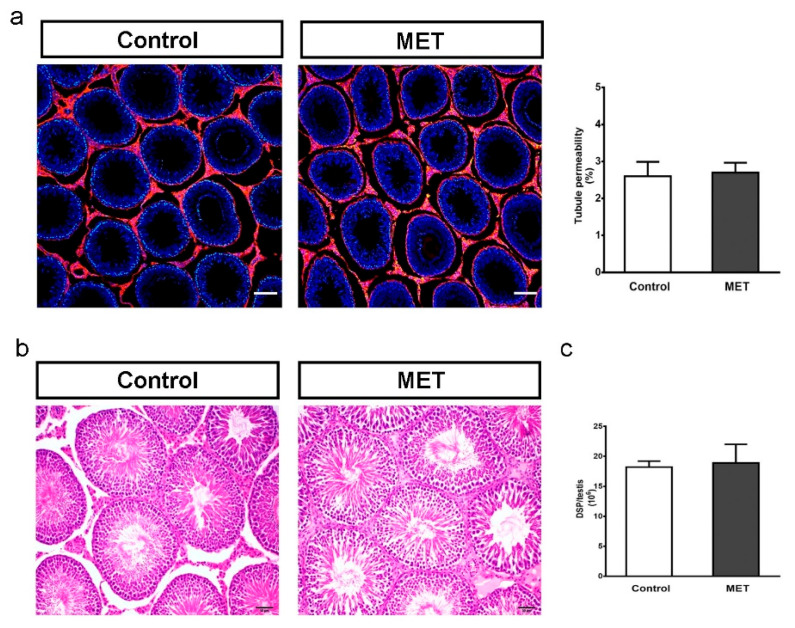
Effect of juvenile metformin treatment on adult BTB integrity, testicular histology, and DSP. Animals from the control and MET groups were treated from Pnd14 to Pnd30. At Pnd90, testes were removed. (**a**) BTB permeability was evaluated. Representative pictures are shown in the left panels (scale bar, 100 μm). The quantification of permeable tubules was performed. Results are expressed as mean ± S.D. (n = 5/group). (**b**) Testes were fixed in Bouin solution. Sections were stained with hematoxylin/eosin and examined by light microscopy. Representative pictures are shown (scale bar, 50 μm). (**c**) Daily sperm production (DSP) was measured in control and MET group samples. Results are expressed as the mean ± S.D. (n = 4/group).

**Table 1 biology-13-00330-t001:** Effect of metformin on Sertoli cell viability.

	Cell Viability (% Basal)
Basal	100
Met 24 h	99 ± 3
Met 48 h	106 ± 5
Met 72 h	105 ± 4

Sertoli cell monolayers were incubated in the absence (Basal) or presence of Met 10 mM for variable periods (24, 48, or 72 h). Then, SC viability was evaluated. Data are expressed as a percentage of the basal condition and presented as mean ± SD of triplicate incubations in one representative experiment out of three.

**Table 2 biology-13-00330-t002:** Animal and testis weight from control and MET groups on Pnd31.

	Control	MET
Animal weight (g)	120.6 ± 12	123.4 ± 5.7
Testis weight (mg)	362 ± 83	372 ± 76

Values represent mean ± S.D. obtained from 5 rats in each experimental group.

**Table 3 biology-13-00330-t003:** Animal, testis, and epididymal weight from control and MET groups on Pnd90.

	Control	MET
Body weight (g)	461.0 ± 8.5	463.3 ± 11.0
Testis weight (g)	1.8 ± 0.2	1.7 ± 0.2
Epididymal weight (g)	0.63 ± 0.05	0.61 ± 0.04

Animals (n = 4/group) were treated with 200 mg/kg/day of MET from Pnd 14 to Pnd30. At Pnd 90, body, testicular, and epididymal weights were assessed. Results are presented as mean ± S.D.

## Data Availability

The data underlying this article will be shared on reasonable request to the corresponding author.

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
