# Peer review of "Effect of Metformin on Sertoli Cell Fatty Acid Metabolism and Blood–Testis Barrier Formation"

_biology, 2024, doi:10.3390/biology13050330_

Round 1

Reviewer 1 Report

Comments and Suggestions for Authors

Rindone and colleagues investigate the effects of metformin on rat Sertoli cell function in culture and on maintenance of spermatogenesis in vivo. The studies have potential impact because metformin is used by diabetics of all ages. The authors provide evidence that metformin treatment of cultured rat Sertoli cells alters expression of proteins required for lactate supply to germ cells and energy production as well as tight junction integrity of Sertoli cells. Metformin treatment of juvenile rats from post-natal days (PND) 15 to 30 caused small changes in blood-testis barrier (BTB) integrity at PND 30. However, any disruptions caused by the metformin treatments were not maintained in adult rats, which did not show altered integrity of the blood-testis barrier (BTB) or disrupted spermatogenesis. Overall, the studies and data interpretation are well performed, but the lack of physiological effects of metformin in vivo may limit the impact of the work.

Specific criticisms:

1.        For the culture studies, the authors employ levels of testosterone (1 micromolar) that are about 10-fold higher than physiological levels. However, the high levels of testosterone likely were not a confounding factor in this study.

2.        Figure 1: With the exception of Cpt1, metformin effects on Sertoli cell genes required for fatty acid oxidation are relatively small (30-50%), which are in line with the relatively small effects of metformin on Sertoli cell function.

3.        Figure 2b: The number of biological replicates for the western blot is not provided. Quantitation of western blots is required.

4.        Line 426 requires clarification.

5.        Figure 5: In comparison to findings reported for Sertoli cell cultures in Figure 3, relatively small (or no) changes are observed in BTB integrity as well as the levels of Ar, Ocln, Cldn11 and ZO-1 mRNA isolated from testes of rats treated with metformin. It is possible that the findings are representative of the mixed cell population of the testes that may obscure small changes in Sertoli cell mRNA expression that occur due to metformin treatment.

6.        Figure 7:  The number of biological replicates is needed.

7.        There does not appear to be data to support the statements made by the authors about metformin causing disruption of the seminiferous tubules in juvenile animals that is later repaired in the adult. For example, BTB integrity is decreased by what appears to be less than 40% in figure 5a and only one of 12 tubules show infiltration of tracer. The data shown does not provide strong support for substantial disruption of juvenile tubules that would need to be repaired. Statements about repair occurring in the adult should be softened or defended with stronger data. The authors might consider placing more emphasis on the valuable information obtained that metformin likely will not cause lasting damage to the testes of diabetics that must take the drug.

Comments on the Quality of English Language

Overall, the manuscript is well written and clear.

Reviewer 2 Report

Comments and Suggestions for Authors

I reviewed with interest this paper on the effects of metformin on Sertoli cell fatty acid metabolism and 

blood-testis barrier formation, particularly during juvenile stages. The theme is an innovative and relevant study area, given the use of this medication in pediatric populations with type 2 diabetes. The focus on Sertoli cells, essential for supporting spermatogenesis, is well chosen. Using both in vitro and in vivo models is appropriate, offering a comprehensive approach to studying the cellular and organismal effects of metformin. However, additional age groups should be included to cover earlier or later stages of testicular development.
